# Light-induced switching between singlet and triplet superconducting states

Steven Gassner [1] ✉, Clara S. Weber [1,2] & Martin Claassen [1] ✉

While the search for topological triplet-pairing superconductivity has remained a challenge, recent developments in optically stabilizing metastable superconducting states suggest a new route to realizing this elusive phase. Here, we devise a testable theory of competing superconducting orders that permits ultrafast switching to an opposite-parity superconducting phase in centrosymmetric crystals with strong spin-orbit coupling. Using both microscopic and phenomenological models, we show that dynamical inversion symmetry breaking with a tailored light pulse can induce odd-parity (spin triplet) order parameter oscillations in a conventional even-parity (spin singlet) superconductor, which when driven strongly can send the system to a competing minimum in its free energy landscape. Our results provide new guiding principles for engineering unconventional electronic phases using light, suggesting a fundamentally non-equilibrium route toward realizing topological superconductivity.

Topological superconductors are elusive unconventional superconducting phases[1-3] that can host topologically-protected Majorana boundary modes and non-Abelian vortex excitations[4,5], which are of fundamental as well as tremendous practical interest as a route towards fault-tolerant quantum computing[6]. Spin-triplet superconductors with finite angular momentum Cooper pairs[7,8] have long been regarded as particularly promising candidates, with degeneracies between nodal order parameters expected to favor a chiral topological superconducting state[9]. However, spin-triplet pairing remains rare in nature and signatures of chiral topological order remain inconclusive, despite several candidate compounds such as $Sr_2RuO_4$[10-12] or $UTe_2$[13,14] being placed under exceptional experimental scrutiny.

At the same time, a series of pioneering pump-probe experiments have established irradiation with light as an alternative and fundamentally non-equilibrium tool for interrogating and manipulating superconducting phases on ultrafast time scales, ranging from time-resolved probes of Higgs[15-18] and Leggett[19-22] mode oscillations in conventional and multi-gap superconductors to the light-induced enhancement or induction of long-lived superconducting signatures in the fullerides[23-25]. With the underlying mechanisms still under substantial debate, these observations coincide with broader

experimental[26-29] and theoretical efforts[30-34] in exploring thermal and non-thermal pathways to suppress or control competing ordered phases with light.

These results immediately raise the tantalizing question of whether elusive topological spin-triplet superconducting states can instead emerge as metastable phases upon irradiating a conventional superconductor with light. Consider an inversion-symmetric material with a conventional s-wave superconducting phase that preempts a closely competing topological spin-triplet pairing instability in equilibrium. In addition to the usual Higgs mode, such a system must necessarily retain additional amplitude modes in alternative pairing channels called Bardasis-Schrieffer (BS) modes[35-37], which include odd-parity amplitude modes corresponding to spin-triplet pairing. These modes can lie below the gap in the case of closely-competing orders[38] provided that long-range Coulomb interactions do not push the mode into the pair-breaking continuum[39]. While this mode remains decoupled in equilibrium from conventional Higgs oscillations in centrosymmetric materials, a tailored ultrafast light pulse can transiently break inversion symmetry and combine with strong spin-orbit coupling to induce odd-parity amplitude oscillations in a conventional superconductor. At a moderate fluence, the material can refuse to relax back to its equilibrium phase, instead becoming

[1]Department of Physics and Astronomy, University of Pennsylvania, Philadelphia, PA 19104, USA. [2]Institut für Theorie der Statistischen Physik, RWTH Aachen and JARA - Fundamentals of Future Information Technology, D-52056 Aachen, Germany. ✉e-mail: sgassner@sas.upenn.edu; claassen@sas.upenn.edu

trapped in a metastable spin-triplet superconducting phase, with thermalization to the equilibrium spin-singlet phase suppressed due to restored inversion symmetry after the pulse.

In this work, we illustrate this mechanism as both a new route towards engineering spin-triplet and topological superconducting phases out of equilibrium as well as a generic probe of competing odd-parity instabilities in conventional superconductors. The protocol is summarized in Fig. 1. We first identify ultrafast and two-color pulses as two complementary routes to dynamically break inversion symmetry, and demonstrate using a minimal model of quasiparticle dynamics that they can conspire with spin-orbit coupling to transiently induce odd-parity order parameter oscillations in a system with even-parity order. We then derive an effective time-dependent Ginzburg-Landau theory dictated via symmetry and present an explicit switching protocol for driving the system to settle into a metastable odd-parity superconducting state. Remarkably, we find that the coupling between equilibrium conventional $s$-wave order and a competing spin-triplet order parameter necessarily scales linearly with the field strength, in contrast to ordinary Higgs mode excitations which scale quadratically with the field. We illustrate the proposed mechanism for light-induced switching to a triplet superconductor by the example of two different lattice models, one of which features a metastable chiral topological superconducting state, and discuss implications for real systems such as dilute-doped 1T′ WTe$_2$. Our results reveal new guiding principles for engineering metastable unconventional superconducting states using light.

## Results

### Dynamical inversion symmetry breaking with light

While conventional centrosymmetric superconductors typically comprise local spin-singlet Cooper pairs with even parity and net spin $S = 0$, spin-triplet superconductivity entails odd-parity pairs with net spin $S = 1$ and three allowed values $m_s = 0, \pm 1$ for the spin component. Spin and parity are linked by necessity; since the combination of inversion and spin-exchange has the net effect of exchanging two fermions, even-parity pairs must be singlets while odd-parity pairs must be triplets. Coupling singlet and triplet orders therefore immediately requires the breaking of both inversion and $SU(2)$ spin rotation symmetries. The latter is readily broken in superconductors

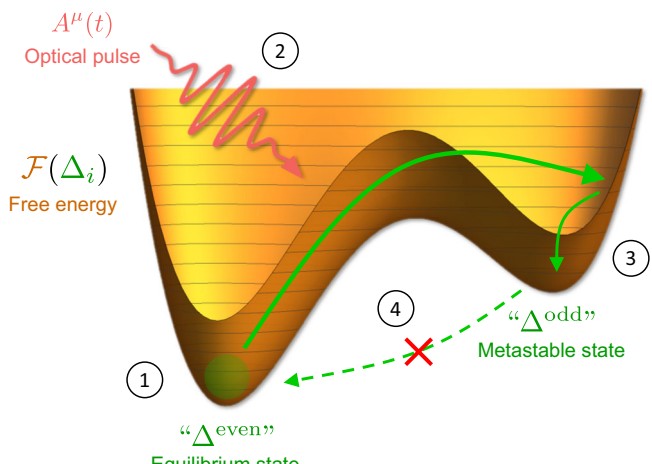

**Fig. 1 | Light-induced metastable triplet superconductors.** (1) The system starts in a superconducting phase that is even-parity (singlet-pairing) in equilibrium. (2) An optical pulse dynamically breaks inversion symmetry, driving the system toward a local minimum in its free energy landscape that comprises multiple competing order parameters. (3) The system relaxes into a metastable superconducting state that is odd-parity (triplet-pairing). (4) Since inversion symmetry is restored after the pulse, equilibration back to the opposite-parity state is suppressed.

with strong spin-orbit coupling such as heavy-fermion compounds. Most notably, monolayer 1T′ WTe$_2$ has been observed to host both a quantum spin hall insulating phase[40–43] due to strong spin-orbit coupling and a proximal superconducting phase for dilute doping[42] with a rich array of predicted competing superconducting states with different pairing symmetries[44], rendering it a prime candidate to search for an out-of-equilibrium topological superconducting phase.

In contrast, inversion-breaking turns out to be more subtle, requiring careful consideration of the optical driving scheme. A simple monochromatic light wave is not sufficient, since it preserves a more general dynamical inversion symmetry defined by a combination of parity and time translation by half the wave period $T$,

$$\mathbf{r} \to -\mathbf{r}, \quad t \to t - T/2. \tag{1}$$

This dynamical symmetry can be strongly broken in two ways: (a) via the envelope of an ultrafast pulse, or (b) via a two-color pulse $\mathbf{E}(t) = \mathcal{E}_1 \cos(\omega_1 t) + \mathcal{E}_2 \cos(\omega_2 t)$ if the two constituent frequencies are not odd harmonics $\omega_1 = p\omega_0$, $\omega_2 = q\omega_0$ of a common frequency $\omega_0$[45–47]. Dynamical symmetries have been extensively studied as a means of high-harmonic generation in atomic and molecular systems[48–50] and are more recently being used for optically controlling solids[51–54].

We first illustrate the ramifications of breaking these two symmetries for a minimal mean-field model of a centrosymmetric honeycomb lattice superconductor

$$\hat{H} = -t \sum_{\langle ij \rangle \sigma} \hat{c}_{i\sigma}^\dagger \hat{c}_{j\sigma} - i\lambda \sum_{\langle\langle ij \rangle\rangle \sigma} \sigma \nu_{ij} \hat{c}_{i\sigma}^\dagger \hat{c}_{j\sigma} \\ + U \sum_i \hat{n}_{i\uparrow} \hat{n}_{i\downarrow} + U' \sum_{\langle ij \rangle \sigma\sigma'} \hat{n}_{i\sigma} \hat{n}_{j\sigma'} \tag{2}$$

with effective attractive local ($U$) and nearest-neighbor ($U'$) interactions. Importantly, the inclusion of spin-orbit coupling $\lambda$ via spin-dependent next-nearest-neighbor hopping with phases $\nu_{ij} = \pm 1$ for left or right turns[55] reduces spin rotation symmetry to $U(1)$, permitting a coupling between singlet and $m_s = 0$ triplet pairs $\sim (\hat{c}_{-\mathbf{k}\uparrow} \hat{c}_{\mathbf{k}\downarrow} \mp \hat{c}_{-\mathbf{k}\downarrow} \hat{c}_{\mathbf{k}\uparrow})/\sqrt{2}$.

A standard BCS mean-field decoupling of the interaction in the Cooper channel introduces the superconducting gap function $\Delta_{\alpha\beta}(\mathbf{k}) = \sum_i f_{\alpha\beta}^i(\mathbf{k}) \Delta_i$ with sublattice indices $\alpha, \beta$ which can be decomposed into pairing channels

$$\Delta_i = \frac{\upsilon_i}{L^d} \sum_{\mathbf{k}} \bar{f}_{\alpha\beta}^i(\mathbf{k}) \langle \hat{c}_{-\mathbf{k}\beta\downarrow} \hat{c}_{\mathbf{k}\alpha\uparrow} \rangle \tag{3}$$

classified in terms of the irreducible representations of the crystal point group with form factors $f_{\alpha\beta}^i(\mathbf{k})$ and channel-projected interactions $\upsilon_i$ that depend on $U, U'$ [see Methods].

Suppose now that a conventional $s$-wave superconducting phase in equilibrium is irradiated with a weak but wide pump pulse, which couples to electrons via the Peierls substitution with a vector potential $\mathbf{A}(t)$ polarized in the $x$ direction. The pulse is parameterized via the dimensionless field strength $|A| = ea_0 \mathcal{E}_0 / \hbar\omega$ where $\mathcal{E}_0$, $\omega$ and $a_0$ denote the electric field amplitude, frequency, and the lattice constant, respectively. For numerical expediency, in Fig. 2 we use effective interactions $U = U' = -2$ and $\beta = 10$, in units of hopping. Strikingly, the onset of $p_x$-wave order parameter oscillations for weak light pulses is linear in the field strength, shown in Fig. 2a–b for a two-color pulse, and in stark contrast to ~$A^2$ scaling for ordinary amplitude mode oscillations. Furthermore, the amplitude of $p_x$ order scales linearly with $\lambda$, completely vanishing in the SU(2) symmetric limit $\lambda = 0$. To illustrate the role of inversion symmetry breaking, Fig. 2c and d depict the order parameter response to a two-color

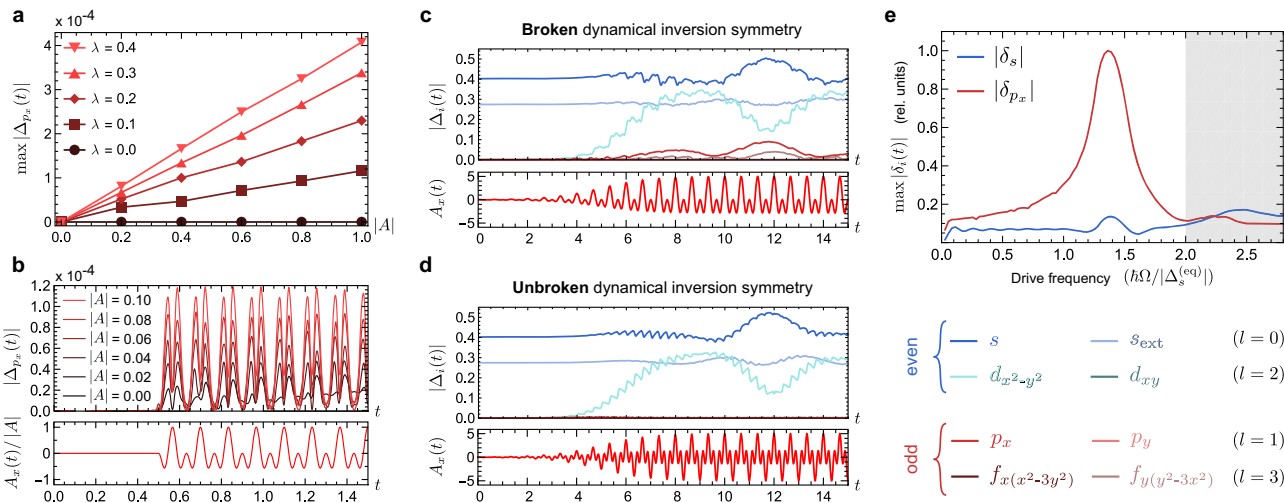

**Fig. 2 | Dynamical inversion symmetry breaking and odd-parity Bardasis-Schrieffer (BS) modes. a, b** Demonstration of a linear-in-$|A|$ and linear-in-$\lambda$ (spin-orbit parameter) coupling between an equilibrium order parameter $\Delta_s$ and a $p_x$-wave order parameter $\Delta_{p_x}$. A two-color $x$-polarized pulse that breaks dynamical inversion symmetry is applied with various small magnitudes $|A|$, and example time-plots of the resulting fluctuations in $|\Delta_{p_x}(t)|$ are plotted in (**b**). The maxima of these fluctuations are plotted versus $|A|$ in (**a**), for five different magnitudes of the spin-orbit coupling strength $\lambda$ in units of hopping. **c** Time dynamics resulting from a two-color pulse that breaks dynamical inversion symmetry (with a 2:1 frequency ratio).

**d** Time dynamics resulting from a two-color pulse that preserves dynamical inversion symmetry (with a 3:1 frequency ratio). Both (**c**) and (**d**) start from purely even superconducting order (indicated by blue curves), and the onset of odd orders (indicated by red and pink curves) is suppressed when dynamical inversion symmetry is unbroken (see legend in bottom right). **e** Short-time deviations in the $s$-wave and $p_x$-wave order parameters for weak driving with $x$-polarized light as a function of frequency $\Omega$, revealing a BS mode for the $p_x$-wave order parameter that is sub-gap (indicated by the unshaded region).

pulse with 2:1 and 3:1 frequency ratio, respectively. A 3:1 frequency ratio preserves a dynamical inversion symmetry [Eq. (1)] with time translation $t \to t + \pi/\omega$; $p$-wave order oscillations are consequently dramatically suppressed at short times. Conversely, broken dynamical inversion symmetry efficiently excites odd-parity order already for short times [Fig. 2c]. Note that, due to the nonlinear nature of the quasiparticle equations of motion, these heuristics only apply for time scales of only a few cycles of the pump pulse. Central to efficient switching to triplet order, Fig. 2e reveals a BS mode resonance for subdominant $p$-wave order parameter oscillations that crucially lies below the ordinary Higgs mode and pair-breaking excitations. Spectroscopic observation of this mode would immediately provide an experimentally accessible handle to probe the existence of a subdominant pairing channel, and would importantly suggest that stronger ultrafast excitation of this mode can potentially nudge the system well beyond linear order parameter oscillations and into a metastable competing phase with triplet pairing.

## Optical switching to a metastable state

Insight into whether strongly driving a triplet BS mode can allow for light-induced switching to a metastable odd-parity superconductor can be readily gleaned from an effective time-dependent Ginzburg-Landau (TDGL) description, which encodes the coupling of multiple order parameters to light and importantly accounts for relaxation. In this picture, a suitably tailored pulse liberates the superconducting order parameter from its global free energy minimum and brings it close enough to a proximal local minimum that it relaxes into a metastable opposite-parity phase. A minimal Lagrangian that describes this process reads

$$\mathcal{L} = -\overline{\Delta}_i \left( \Gamma_{ij}^{(2)} \partial_t^2 + \Gamma_{ij}^{(1)} \partial_t \right) \Delta_j - \beta \mathcal{F}(\Delta_i, \boldsymbol{\nabla}\Delta_i, \beta). \quad (4)$$

and includes a kinetic contribution with damping coefficients $\Gamma_{ij}^{(1)}$ and inertial coefficients $\Gamma_{ij}^{(2)}$. The equilibrium free energy $\mathcal{F}$ is taken to generalize the usual Ginzburg-Landau action to $N$ order parameters that crucially include subdominant orders not stabilized in

equilibrium, and formally reads

$$\begin{aligned} \beta\mathcal{F} &= \mathcal{A}_{ij}\overline{\Delta}_i\Delta_j + \frac{1}{2}\mathcal{B}_{ijmn}\overline{\Delta}_i\Delta_j\overline{\Delta}_m\Delta_n \\ &+ \mathcal{C}_{ij}^\mu\overline{\Delta}_i\nabla^\mu\Delta_j + \mathcal{D}_{ij}^{\mu\nu}\nabla^\mu\overline{\Delta}_i\nabla^\nu\Delta_j. \end{aligned} \quad (5)$$

Here, the bar denotes complex conjugation, and summation over repeated indices is implied. Coefficients $\mathcal{A}_{ij}$, $\mathcal{B}_{ijmn}$, and $\mathcal{D}_{ij}^{\mu\nu}$ are tensorial generalizations of the usual Ginzburg-Landau coefficients for quadratic, quartic, and gradient contributions. We use subscript Latin indices $(i, j, m, n = 1, ..., N)$ to index different order parameters, and superscript Greek indices $(\mu, \nu)$ to index spatial directions. The theory accurately represents a multi-dimensional free energy landscape in the vicinity of the critical temperature $T_c$ for the equilibrium even-parity instability. Finally, we couple the superconductor to light by introducing minimal coupling to a gauge field in velocity gauge, $\nabla^\mu \longrightarrow \nabla^\mu + i\frac{2e}{\hbar}A^\mu(t)$, where $-e < 0$ is the electron charge. We discard subsequent gradient terms by considering only spatially homogeneous irradiation and order parameters ($\boldsymbol{\nabla}\Delta_i = 0$). From this, one can derive Euler-Lagrange equations of motion $\left( \frac{\partial\mathcal{L}}{\partial\overline{\Delta}_i} - \partial_t \frac{\partial\mathcal{L}}{\partial(\partial_t\overline{\Delta}_i)} = 0 \right)$ that describe light-induced dynamics of competing orders.

The structure of the TDGL action is dictated solely by parity and angular momentum conservation, and importantly permits a coupling between even- and odd-parity order parameters already to linear order in $A^\mu(t)$. If the lattice has $C_n$ rotational symmetry with $n \geq 2$, order parameters for $s$-, $p$-, $d$-wave instabilities can be enumerated by their angular momentum eigenvalues $e^{i2\pi l/n}$, with $l = 0$, $l = \pm 1$, $l = \pm 2$ (modulo $n$) respectively. An appealing selection rule permits light-induced coupling $\mathcal{C}_{ij}^\mu$ between superconducting orders $i, j$ with $\Delta l = \pm 1$ at linear order in the field. Notably, this dictates that the Lagrangian couples an equilibrium $s$-wave order parameter to odd-parity $p$-wave order already at linear order in $A^\mu(t)$, in agreement with the quasiparticle dynamics of Fig. 2a. Conversely, second order in $A^\mu(t)$ contributions couple same-parity order parameters with $\Delta l = 0$ or $\Delta l = \pm 2$, capturing

the excitation of the conventional amplitude mode. This observation has intriguing consequences for Higgs mode experiments (see Summary and outlook).

To address the existence of a metastable triplet superconductor as a local minimum of the free energy landscape, we explicitly compute the multi-component Ginzburg-Landau coefficients for microscopic Hamiltonians, thus connecting the top-down phenomenological approach to the bottom-up microscopic approach of the previous section. Starting from a generic multiband Hamiltonian with effective attractive pairing interactions, the generalized Ginzburg-

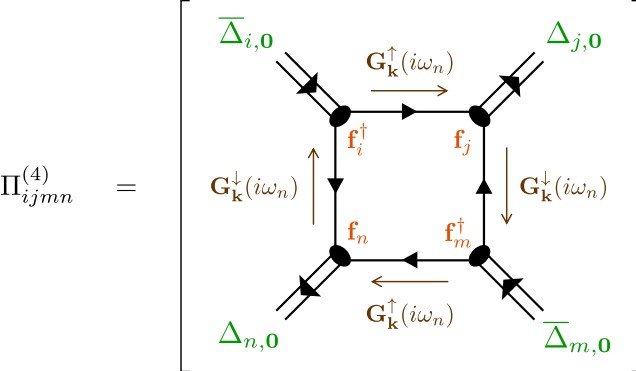

$$\Pi_{ij}^{(2)}(\mathbf{q}) = \left[ \overline{\Delta}_{i,\mathbf{q}} \quad \mathbf{f}_i^\dagger \quad \mathbf{f}_j \quad \Delta_{j,-\mathbf{q}} \right]$$

with $\mathbf{G}_{\mathbf{k}+\frac{\mathbf{q}}{2}}^\uparrow(i\omega_n)$ and $\mathbf{G}_{\mathbf{k}-\frac{\mathbf{q}}{2}}^\downarrow(i\omega_n)$

$$\Pi_{ijmn}^{(4)} =$$

**Fig. 3 | Effective action via diagrams.** Diagrammatic representation of Eqs. (7) and (8), which summarize how to calculate the generalized Ginzburg-Landau coefficients as an effective field theory starting from a microscopic Hamiltonian.

Landau coefficients can be computed diagrammatically as

$$\mathcal{A}_{ij} = -\frac{1}{\nu_{(i)}}\delta_{ij} + \Pi_{ij}^{(2)}(\mathbf{0}), \quad \mathcal{B}_{ijmn} = \frac{1}{2}\Pi_{ijmn}^{(4)}, \quad (6)$$

with correlation functions

$$\Pi_{ij}^{(2)}(\mathbf{q}) = \text{Tr}\left\{ \mathbf{f}_{i,\mathbf{k}}^\dagger \mathbf{G}_{\mathbf{k}-\frac{\mathbf{q}}{2}}^\uparrow \mathbf{f}_{j,\mathbf{k}} \mathbf{G}_{\mathbf{k}+\frac{\mathbf{q}}{2}}^\downarrow \right\}, \quad (7)$$

$$\Pi_{ijmn}^{(4)} = \text{Tr}\left\{ \mathbf{f}_{i,\mathbf{k}}^\dagger \mathbf{G}_{\mathbf{k}}^\uparrow \mathbf{f}_{j,\mathbf{k}} \mathbf{G}_{\mathbf{k}}^\downarrow \mathbf{f}_{m,\mathbf{k}}^\dagger \mathbf{G}_{\mathbf{k}}^\uparrow \mathbf{f}_{n,\mathbf{k}} \mathbf{G}_{\mathbf{k}}^\downarrow \right\}, \quad (8)$$

depicted in Fig. 3, and Matsubara Green's functions for particles and holes given by

$$\mathbf{G}_{\mathbf{k}}^\uparrow(i\omega_n) = \frac{1}{i\omega_n - \mathbf{h}_{\mathbf{k}\uparrow}}, \quad \mathbf{G}_{\mathbf{k}}^\downarrow(i\omega_n) = \frac{1}{i\omega_n + \mathbf{h}_{-\mathbf{k}\downarrow}^\top}. \quad (9)$$

In the above equations, $\text{Tr} \equiv \frac{1}{\beta L^d}\sum_{\mathbf{k},\omega_n} \text{tr}$, indicating a sum over momenta, orbital indices, and Matsubara frequencies. The gradient terms $\mathcal{C}_{ij}^\mu$ and $\mathcal{D}_{ij}^{\mu\nu}$ are calculated from expanding $\Pi_{ij}^{(2)}(\mathbf{q})$ in powers of $\mathbf{q}$.

We first demonstrate key qualitative features by example of the conceptually simplest single-band model on a rectangular lattice (Fig. 4a) that captures the essential physics for switching to metastable odd-parity superconductor. We choose a hopping anisotropy $\delta t = 0.2$, chemical potential $\mu = -0.1$, and interaction parameters $\nu_s = -1.75$, $\nu_{p_x} = \nu_{p_y} = -2.5$, in units of hopping. As spin-orbit interactions in centrosymmetric crystals require at least a two-orbital unit cell, we break $SU(2)$ via a small Zeeman magnetic field $\Delta_{\text{Zeeman}} = 0.1$ [see "Methods"]. The free energy parameters are computed from the microscopic model, and the phenomenological inertial/damping coefficients are set to $\gamma = 1.0$, $\eta = 0.1$. The temperature $T = 0.1$ lies below the critical temperature for the $s$-wave and $p_x$-wave channel, with $s$-wave pairing stabilized in equilibrium. Crucially, the coupling between these channels is first-order in $\mathbf{A}(t)$. For simplicity, we assume that pairing interactions in the $d$-wave and extended $s$-wave channels vanish. Furthermore, an $x$-polarized pulse couples solely to $p_x$ order, yielding

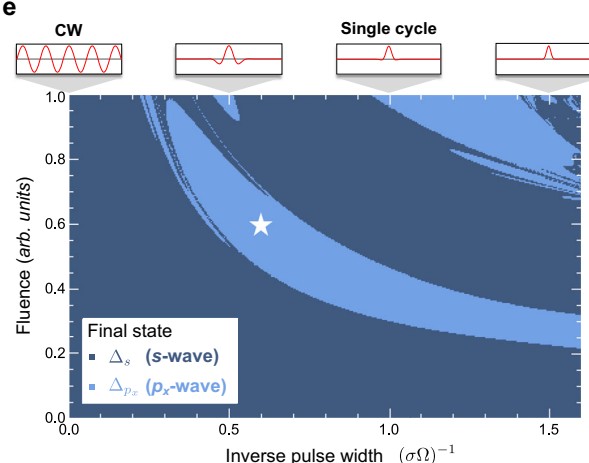

**Fig. 4 | Engineering metastable triplet superconductors with light. a** Schematic of a minimal rectangular tight-binding model of competing singlet and triplet pairing instabilities. **b** Free energy landscape plot of $\text{Re}(\Delta_{p_x})$ vs $\text{Re}(\Delta_s)$, with the time-dependent trajectory in (**c**) plotted in green. **c** Time plots of $A_x(t)$ and $|\Delta_i(t)|$ for $s_{\text{total}}$ [Methods], $p_x$, and $p_y$ order parameters subjected to an example Gaussian pulse. **d** Schematic plot showing the spectral profile of the pulse in (**c**) in relation to

the Bardasis-Schrieffer mode frequency $\Omega_{\text{BS}}$ and the edge of the quasiparticle continuum (QPC). **e** A parametric plot of fluence versus inverse pulse width showing the final state induced by a family of Gaussian pulses, with frequency $\Omega = 0.75\,\Omega_{\text{BS}}$ centered slightly below the BS mode frequency. A star indicates the pulse parameters used in **b**–**d**.

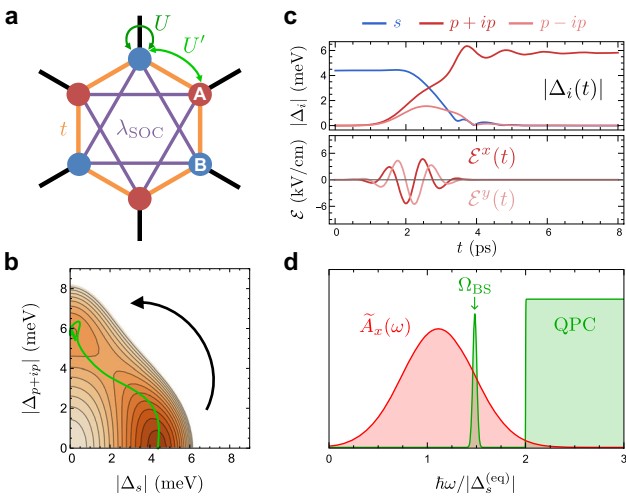

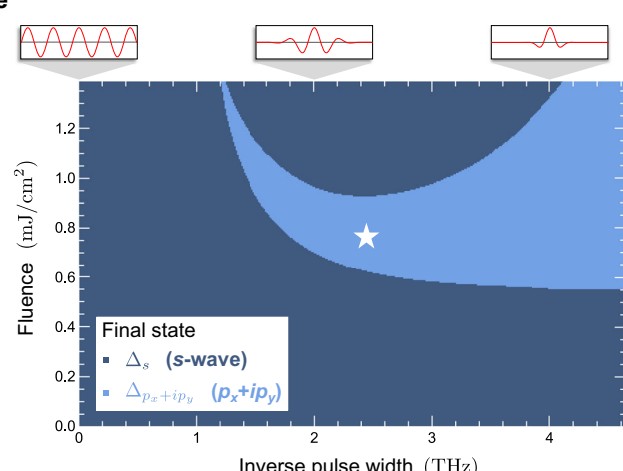

**Fig. 5 | Switching between trivial *s* and topological triplet *p ± ip* pairing in honeycomb superconductors. a** Schematic of a minimal honeycomb-lattice conventional *s*-wave superconductor with a competing chiral triplet pairing instability. **b** Order parameter trajectory in the free energy landscape as a function of the magnitudes of the *s* and *p + ip* order parameters, demonstrating the stable switching of the equilibrium order parameter to a metastable chiral state. A small difference in the relaxation rates for the two chiral states is assumed. **c** Dynamics of $A^x(t)$, $A^y(t)$, and $|\Delta_i(t)|$ for $\Delta_s$ and $\Delta_{p\pm ip}$. **d** Schematic plot of the

spectral profile of the pulse in (**c**) showing strong overlap with the BS mode frequency $\Omega_{BS}$ and minimal overlap with the quasiparticle continuum (QPC). **e** Parametric plot in terms of inverse pulse width (in THz) and fluence (in mJ/cm²) showing which Gaussian pulses centered at $\Omega = 0.75\,\Omega_{BS}$ lead to successful switching to the metastable $p + ip$ state. The window is chosen so that no more than 10% of the fluence overlaps with the QPC. A star indicates the pulse parameters used in **b**–**d**.

an appealingly simple minimal action

$$
\begin{aligned}
\mathcal{L} = \sum_{i=s,p} & \left[ \overline{\Delta}_i \left( \gamma \partial_t^2 + \eta \partial_t \right) \Delta_i + a_i |\Delta_i|^2 + b_i |\Delta_i|^4 \right] \\
& + \sum_{i=s,p} \left( \nabla^\mu - i\frac{2e}{\hbar} A^\mu \right) \overline{\Delta}_i \left( \nabla^\mu + i\frac{2e}{\hbar} A^\mu \right) \Delta_i \\
& + \frac{1}{2} b_{sp} \left( 4|\Delta_s|^2 |\Delta_p|^2 + \overline{\Delta}_s^2 \Delta_p^2 + \overline{\Delta}_p^2 \Delta_s^2 \right) \\
& + \left[ c_{sp} \overline{\Delta}_p \left( \nabla^x + i\frac{2e}{\hbar} A^x \right) \Delta_s + \text{c.c.} \right],
\end{aligned}
\tag{10}
$$

where we abbreviate $\mathcal{A}_{ii} \equiv a_i$, $\mathcal{B}_{iiii} \equiv b_i$, and $\mathcal{B}_{iijj} \equiv b_{ij}$, and assume equal inertial ($\gamma$) and damping ($\eta$) coefficients without loss of generality. The first two lines describe decoupled TDGL actions for *s* and $p \equiv p_x$ order parameters. The third line describes their quartic coupling that dictates the emergence of a metastable minimum, and the last line couples *s* and *p* orders to linear order in the light field $A^\mu$. In the rectangular model, this coupling keeps the *s*- and $p_x$-wave order parameters completely real, allowing the dynamics to be completely captured by a visualizable two-dimensional free energy contour plot (see Fig. 4b).

Intriguingly, as a function of pulse width and fluence, one finds a contiguous parametric family of Gaussian pulses that result in the final state settling into the pure $\Delta_{p_x}$ stationary point of the free energy, as in Fig. 4e. The presence of a clear threshold for the width of the Gaussian pulse is evidence that dynamical inversion symmetry breaking, which occurs more strongly with a tighter pulse width, is crucial for efficient coupling between opposite-parity orders. We find switching to be most reliable for strong driving near the Bardasis-Schrieffer mode frequency (Fig. 2d), which reads

$$
\Omega_{BS} = \sqrt{\frac{1}{\gamma_p} \left( a_p - 3a_s \frac{b_{sp}}{b_s} \right)},
\tag{11}
$$

and is determined by deviations from the global free energy minimum in the direction of the target instability [see Methods]. Light polarized in the *x*-direction drives the order parameter strongly in the direction of a closely competing $p_x$ minimum in its free energy landscape, and

for Gaussian pulses centered at this frequency with widths on the order of a few resonant periods (see Fig. 4e), the system is efficiently switched to the target metastable state. Figure 4d and e together reveal a trade-off whereby the pulse needs to be short enough to allow for strong dynamical inversion symmetry breaking while being wide enough (i.e. spectrally sufficiently narrow) to avoid excitation of quasiparticles across the gap (excitations that are not captured by TDGL). The latter constraint leads us to detune the central frequency of the Gaussian pulse to be slightly below $\Omega_{BS}$, decreasing the spectral overlap with the quasiparticle continuum while still maintaining a large overlap with the BS mode. [See the Supplementary Material for further details.]

We now turn to a realistic model of a conventional centrosymmetric superconductor with strong spin-orbit coupling on the honeycomb lattice, with a putative closely-competing triplet pairing instability (Fig. 5a). This models the low-energy physics near the Dirac points of several systems of recent interest, including kagome superconductors[56–58], honeycomb materials with large spin-orbit coupling and recently-reported superconductivity[59–61], and many moiré heterostructures of transition metal dichalcogenides[62–66]. Rotation symmetry and energetics dictate that the chiral triplet states $\Delta_{p\pm ip} \equiv \frac{1}{\sqrt{2}} \left( \Delta_{p_x} \pm i\Delta_{p_y} \right)$ are stable local free energy minima, while the nodal $\Delta_{p_x}$ and $\Delta_{p_y}$ states individually are not. Figure 5c depicts the free energy landscape with an equilibrium *s*-wave minimum and proximal $p \pm ip$ local minima. This is therefore a promising example of a truly metastable non-equilibrium chiral superconducting state in a system with conventional *s*-wave order in equilibrium.

The key parameter allowing singlet-triplet switching at first order in the gauge field $\mathbf{A}(t)$ is the coefficient $\mathcal{C}_{ij}^\mu$ in Eq. (5). For the honeycomb model, up to first order in the spin-orbit coupling strength $\lambda$, this matrix element

$$
\begin{aligned}
\mathcal{C}_{ij}^\mu \approx \frac{i\lambda}{2} \int \frac{d^2 \mathbf{k}}{(2\pi)^2} \sum_{mn} & \frac{n_F(\xi_{n\mathbf{k}}) - n_F(-\xi_{m\mathbf{k}})}{\xi_{n\mathbf{k}} + \xi_{m\mathbf{k}}} \\
& \times \partial_\lambda \left( \left\{ \mathcal{A}_{\mathbf{k}}^\mu, \mathbf{f}_{i,\mathbf{k}}^\dagger \right\}_{nm} \left( \mathbf{f}_{j,\mathbf{k}} \right)_{mn} \right. \\
& \left. - \left( \mathbf{f}_{i,\mathbf{k}}^\dagger \right)_{nm} \left\{ \mathcal{A}_{\mathbf{k}}^\mu, \mathbf{f}_{j,\mathbf{k}} \right\}_{mn} \right) \Big|_{\lambda \to 0} + \mathcal{O}(\lambda^2),
\end{aligned}
\tag{12}
$$

becomes a function of the (matrix-valued) non-Abelian Berry connection $\mathcal{A}_{\mathbf{k}mn}^{\mu} \equiv i\langle u_{m\mathbf{k}}|\partial_{k_{\mu}}u_{n\mathbf{k}}\rangle$. Since the energy eigenvalues of the Hamiltonian are at order $\lambda^2$ and higher, the dominant contribution to $\mathcal{C}^{\mu}$ for small $\lambda$ is therefore purely geometric in nature, controlled by the Berry connection. Notably, this effect is distinct from quantum-geometric corrections to the superfluid weight[67,68], and can be equivalently expressed via the Fermi surface Berry connection at low temperatures (see "Methods").

To approximate the energetics in conventional but strongly spin-orbit coupled superconducting systems,[69,70] we choose parameters so that the nearest neighbor hopping amplitude is 1 eV, the spin-orbit strength is 0.1 eV, the system is hole-doped with chemical potential $\mu = -0.4$ eV and the critical temperatures of the $s$ and $p \pm ip$ states are ~10 K for effective renormalized interactions $v_s = -1.25$ eV, $v_p = -2.3$ eV. In this scenario, the superconducting gaps are between $1-10$ meV. The inertial coefficients $\Gamma_{ij}^{(2)}$ in Eq. (4) are set so that the frequency of small amplitude oscillations of $\Delta_i$ matches the expected Higgs mode frequency, $\Omega_i = 2|\Delta_i^{(eq)}|/\hbar$. This sets the time scale of interest to be on the order of picoseconds, with relevant frequencies between $1-10$ THz. The time scale for relaxation (parametrized by $\Gamma_{ij}^{(1)}$ in Eq. (4)) is longer than the Higgs mode period by the dimensionless factor $\beta|\Delta_i^{(eq)}|$[31], which in our case is about 10. Lastly, assuming a lattice constant on the order of Angstroms, our results call for peak field strengths on the order of ~100 kV/cm for ultrafast pulses.

Figure 5 e reveals a broad region of pulse widths (~0.1 ps) and fluences (~1.0 mJ/cm²) that lead to successful switching from $s$ to a chiral triplet $p + ip$ state via an ultrafast circularly polarized pulse. The resulting order parameter trajectory in a subspace of the free energy landscape is shown in Fig. 5c and illustrates key features of the switching protocol, whereby the dynamical inversion symmetry breaking kicks the order parameter in the vicinity of a chiral instability. To allow the system to settle into one of the chiral $p \pm ip$ states within TDGL theory, we set a slight difference in their relaxation rates ($\eta_{\pm} = (1 \pm \delta)\eta$ with $\delta = 0.5$). Though introduced as a phenomenological parameter, one can expect this to arise microscopically from pump-induced time-reversal symmetry breaking of the environmental degrees of freedom that provide dissipation, an effect which is not captured by TDGL theory. The chiral states are stable against fluctuations in the amplitudes and relative phases of all competing order parameters that are supported by nearest-neighbor pairing interactions, providing a promising proof-of-concept for engineering a metastable topological superconducting state.

### Summary and outlook

We present light-induced dynamical inversion symmetry breaking as a generic route to explore competing triplet pairing instabilities in conventional superconductors with strong spin-orbit coupling. Identifying a sub-gap odd-parity BS mode that couples linearly to light, we find that driving this mode with a tailored two-color or ultrafast light pulse can switch the system to a metastable unconventional superconducting state. We illustrate this mechanism for minimal models of superconductors with closely competing pairing instabilities.

Immediate future directions include searching for experimental signatures of a sub-gap opposite-parity Bardasis-Schrieffer mode, in addition to applying this methodology to realistic tight-binding models of candidate materials, including 1T' WTe₂ and moiré heterostructures of transition metal dichalcogenides. Following experiments that use pump-probe microscopy to detect the conventional Higgs mode via a linear scaling between pump intensity and the amplitude of resulting gap fluctuations[16–18], a simple optical protocol may be to search for gap fluctuations scaling with the square root of the pump intensity, indicating an amplitude mode coupling linearly to light. Additionally, an intriguing open problem is

under what general conditions does a subleading stationary point in the free energy become a local minimum. We have found that the only examples of metastable states are Kramers partners with chiral pairing, i.e. $p_x \pm ip_y$ order parameters in lattices where the $p_x$ and $p_y$ orders are degenerate. It would be fruitful to further understand the necessary conditions for a system to support a metastable superconducting instability. Another interesting direction would be to model heating effects from irradiation with an ultrafast pulse, accounting for microscopic mechanisms for heat dissipation not included in our methods. Lastly, throughout this work we use effective renormalized interaction strengths $v_i$ as phenomenological parameters for emergent pairing at low energy scales. Detailed modeling of these parameters in candidate materials, via renormalization group treatments and incorporating RPA effects[39], is an important direction for future work.

## Methods

### Gap function equations of motion via quasiparticle dynamics

Quasiparticle equations of motion for a superconductor coupled to light are derived for a generic BCS-like Hamiltonian

$$
\hat{H} = \sum_{\mathbf{k}\alpha\beta\sigma} \hat{c}_{\mathbf{k}\alpha\sigma}^{\dagger} h_{\alpha\beta}^{\sigma}(\mathbf{k}) \hat{c}_{\mathbf{k}\beta\sigma} \\
+ \frac{1}{L^d} \sum_{\mathbf{k}\mathbf{k}'\alpha\alpha'\beta\beta'} V_{\alpha\beta\alpha'\beta'}(\mathbf{k},\mathbf{k}') \hat{c}_{\mathbf{k}\alpha\uparrow}^{\dagger} \hat{c}_{-\mathbf{k}\beta\downarrow}^{\dagger} \hat{c}_{-\mathbf{k}'\beta'\downarrow} \hat{c}_{\mathbf{k}'\alpha'\uparrow},
\tag{13}
$$

where $\hat{c}_{\mathbf{k}\alpha\sigma}^{\dagger}$ creates an electron in the state with momentum $\mathbf{k}$, orbital index $\alpha \in \{1, ..., N\}$, and spin $\sigma \in \{\uparrow, \downarrow\}$. The dependence of the one-body Hamiltonian $h_{\alpha\beta}^{\sigma}(\mathbf{k})$ on spin encodes spin-orbit coupling that reduces the $SU(2)$ spin rotation symmetry to spin-$z$ conservation. $L$ denotes the linear system size. We decompose the four-fermion interaction vertex $V_{\alpha\beta\alpha'\beta'}(\mathbf{k},\mathbf{k}')$ in terms of a set of orthonormal basis functions $\{f_{\alpha\beta}^i(\mathbf{k})\}$,

$$
V_{\alpha\beta\alpha'\beta'}(\mathbf{k},\mathbf{k}') = \sum_i v_i f_{\alpha\beta}^i(\mathbf{k})\overline{f}_{\alpha'\beta'}^i(\mathbf{k}').
\tag{14}
$$

A standard mean-field decoupling of the interaction in the Cooper channel introduces the superconducting gap function

$$
\Delta_{\alpha\beta}(\mathbf{k}) = \frac{1}{L^d} \sum_{\mathbf{k}'\alpha'\beta'} V_{\alpha\beta\alpha'\beta'}(\mathbf{k},\mathbf{k}') \left\langle \hat{c}_{-\mathbf{k}'\beta'\downarrow}\hat{c}_{\mathbf{k}'\alpha'\uparrow} \right\rangle \\
\stackrel{\text{def}}{=} \sum_i \Delta_i f_{\alpha\beta}^i(\mathbf{k}),
\tag{15}
$$

where the components $\Delta_i$ are classified in terms of the irreducible representations of the crystal point group. The mean-field decomposition of Eq. (13) can be written in Bogoliubov-de Gennes (BdG) form as

$$
\hat{H} = \sum_{\mathbf{k}} \overline{\Psi}_{\mathbf{k}} \mathcal{H}_{\mathbf{k}}^{\text{BdG}} \Psi_{\mathbf{k}} - \sum_i \frac{L^d}{v_i}|\Delta_i|^2,
\tag{16}
$$

where

$$
\Psi_{\mathbf{k}} \stackrel{\text{def}}{=} \begin{pmatrix} \hat{c}_{\mathbf{k},1,\uparrow} \\ \vdots \\ \hat{c}_{\mathbf{k},N,\uparrow} \\ \hat{c}_{-\mathbf{k},1,\downarrow}^{\dagger} \\ \vdots \\ \hat{c}_{-\mathbf{k},N,\downarrow}^{\dagger} \end{pmatrix}, \quad \mathcal{H}_{\mathbf{k}}^{\text{BdG}} = \begin{pmatrix} \mathbf{h}_{\mathbf{k}\uparrow} & \boldsymbol{\Delta}_{\mathbf{k}} \\ \boldsymbol{\Delta}_{\mathbf{k}}^{\dagger} & -\mathbf{h}_{-\mathbf{k}\downarrow}^{\top} \end{pmatrix}.
\tag{17}
$$

The equal-time Nambu Green's function

$$\mathcal{G}_{\mathbf{k}}^{\mathrm{BdG}} \overset{\mathrm{def}}{=} \langle \overline{\boldsymbol{\Psi}}_{\mathbf{k}} \otimes \boldsymbol{\Psi}_{\mathbf{k}} \rangle$$

$$\equiv \begin{pmatrix} \left[\langle \hat{c}_{\mathbf{k}\beta\uparrow}^{\dagger} \hat{c}_{\mathbf{k}\alpha\uparrow} \rangle\right] & \left[\langle \hat{c}_{-\mathbf{k}\beta\downarrow} \hat{c}_{\mathbf{k}\alpha\uparrow} \rangle\right] \\ \left[\langle \hat{c}_{\mathbf{k}\beta\uparrow}^{\dagger} \hat{c}_{-\mathbf{k}\alpha\downarrow}^{\dagger} \rangle\right] & \left[\langle \hat{c}_{-\mathbf{k}\beta\downarrow} \hat{c}_{-\mathbf{k}\alpha\downarrow}^{\dagger} \rangle\right] \end{pmatrix}. \quad (18)$$

obeys equations of motion (setting $\hbar = 1$)

$$i\partial_t \mathcal{G}_{\mathbf{k}}^{\mathrm{BdG}} = \left[\mathcal{H}_{\mathbf{k}}^{\mathrm{BdG}}, \mathcal{G}_{\mathbf{k}}^{\mathrm{BdG}}\right]. \quad (19)$$

$$\mathcal{G}_{\mathbf{k}}^{\mathrm{BdG}}\big|_{t\to-\infty} = n_{\mathrm{F-D}}\left(\mathcal{H}_{\mathbf{k}}^{\mathrm{BdG}}\right), \quad (20)$$

where $n_{\mathrm{F-D}}(\mathbf{X}) \equiv (1 + \exp(\beta\mathbf{X}))^{-1}$ is the Fermi-Dirac distribution function. Combined with the instantaneous self-consistency equation for order parameters, this yields equations of motion for the gap function in terms of the Heisenberg dynamics of the Bogoliubov quasiparticles.

## Symmetry analysis of the Lagrangian

Suppose we demand that the Lagrangian be invariant under some symmetry transformation on the order parameters $\Delta_i$ and the gauge field $A^\mu(t)$,

$$\begin{aligned} \Delta_j &\longmapsto \mathcal{D}_{jj'} \Delta_{j'} \\ A^\mu &\longmapsto \mathcal{R}^{\mu\mu'} A^{\mu'}, \end{aligned} \quad (21)$$

for tensors $\mathcal{D}$ and $\mathcal{R}$. Let $\mathcal{R}$ be orthogonal $((\mathcal{R}^\top)^{\mu\nu}\mathcal{R}^{\nu\rho} = \delta^{\mu\rho})$ and choose a basis for $\Delta_i$ such that $\mathcal{D}$ is diagonal $(\mathcal{D}_{ij} \equiv d_{(i)}\delta_{ij})$. One can then show that,

$$\begin{aligned} \mathcal{R}^{\mu\mu'}\mathcal{N}_{ij}^{\mu'} &= \overline{d}_{(i)}d_{(j)}\mathcal{N}_{ij}^\mu, \\ \mathcal{R}^{\mu\mu'}\mathcal{M}_{ij}^{\mu'\nu'}(\mathcal{R}^\top)^{\nu'\nu} &= \overline{d}_{(i)}d_{(j)}\mathcal{M}_{ij}^{\mu\nu}. \end{aligned} \quad (22)$$

$\mathcal{N}_{ij}^\mu$ is nonzero if and only if $\overline{d}_{(i)}d_{(j)}$ is an eigenvalue of $\mathcal{R}$, and $\mathcal{M}_{ij}^{\mu\nu}$ is nonzero if $\overline{d}_{(i)}d_{(j)}$ is a product of two (not necessarily distinct) eigenvalues of $\mathcal{R}$. For example, if one considers $C_n$ rotational symmetry on a lattice with $n \geq 2$, the constraints given by Eq. (22) yield the $l$ selection rules discussed in Results.

## Effective action from the path integral

We start with the partition function written in terms of an imaginary-time path integral over the Grassmann fields $\overline{\psi}, \psi$,

$$\mathcal{Z} = \int \mathcal{D}[\overline{\psi}, \psi] e^{-(S_0[\overline{\psi},\psi] + S_{\mathrm{int}}[\overline{\psi},\psi])}, \quad (23)$$

with a free-electron action given by,

$$S_0[\overline{\psi}, \psi] = \int_0^\beta d\tau \sum_{\mathbf{k}\sigma\alpha\beta} \overline{\psi}_{\mathbf{k}\alpha\sigma} \left[\delta_{\alpha\beta}\partial_\tau + h_{\alpha\beta}^\sigma(\mathbf{k})\right]\psi_{\mathbf{k}\beta\sigma}, \quad (24)$$

and an interacting action given by,

$$S_{\mathrm{int}}[\overline{\psi}, \psi] = \int_0^\beta d\tau \frac{1}{L^d} \sum_{\mathbf{k}\mathbf{k}'\mathbf{q}\alpha\alpha'\beta\beta'i} v_i f_{\alpha\beta}^i(\mathbf{k})\overline{f}_{\alpha'\beta'}^i(\mathbf{k}') \\ \times \overline{\psi}_{\mathbf{k}+\frac{\mathbf{q}}{2},\alpha\uparrow}\overline{\psi}_{-\mathbf{k}+\frac{\mathbf{q}}{2},\beta\downarrow}\psi_{-\mathbf{k}'+\frac{\mathbf{q}}{2},\beta'\downarrow}\psi_{\mathbf{k}'+\frac{\mathbf{q}}{2},\alpha'\uparrow}. \quad (25)$$

As before, $\alpha, \beta$ denote orbital indices, $\sigma$ denotes a spin index, and $v_i$ denotes the interaction decomposed into pairing channels. A Hubbard-Stratonovich transformation[71] decouples $S_{\mathrm{int}}$ in the Cooper channel in terms of auxiliary fields $\overline{\Delta}_{i,\mathbf{q}}, \Delta_{i,-\mathbf{q}}$,

$$\tilde{S}[\overline{\boldsymbol{\Psi}}, \boldsymbol{\Psi}, \overline{\Delta}, \Delta] = \sum_{\mathbf{k}\mathbf{k}'\omega_n} \overline{\boldsymbol{\Psi}}_{\mathbf{k},\omega_n} \left(\mathcal{G}^{-1}\right)_{\mathbf{k},\mathbf{k}'}^{\omega_n} \boldsymbol{\Psi}_{\mathbf{k}',\omega_n} - \sum_{i\mathbf{q}} \overline{\Delta}_{i,\mathbf{q}} \frac{\beta L^d}{v_i} \Delta_{i,-\mathbf{q}}, \quad (26)$$

where the Nambu spinor $\boldsymbol{\Psi}$ and Gor'kov Green's operator $\mathcal{G}^{-1}$ are defined in terms of their momentum components and Matsubara frequency dependence as follows,

$$\boldsymbol{\Psi}_{\mathbf{k}} \overset{\mathrm{def}}{=} \begin{pmatrix} \psi_{\mathbf{k},1,\uparrow} \\ \vdots \\ \psi_{\mathbf{k},N,\uparrow} \\ \overline{\psi}_{-\mathbf{k},1,\downarrow} \\ \vdots \\ \overline{\psi}_{-\mathbf{k},N,\downarrow} \end{pmatrix}, \quad (27)$$

$$\left(\mathcal{G}^{-1}\right)_{\mathbf{k}+\frac{\mathbf{q}}{2},\mathbf{k}-\frac{\mathbf{q}}{2}}^{\omega_n} = \begin{pmatrix} \left(-i\omega_n + \hat{h}_{\mathbf{k}\uparrow}\right)\delta_{\mathbf{q},\mathbf{0}} & \sum_i \Delta_{i,-\mathbf{q}}\hat{f}_{\mathbf{k},i} \\ \sum_i \overline{\Delta}_{i,-\mathbf{q}}\hat{f}_{\mathbf{k},i}^\dagger & \left(-i\omega_n - \hat{h}_{-\mathbf{k}\downarrow}^\top\right)\delta_{\mathbf{q},\mathbf{0}} \end{pmatrix}. \quad (28)$$

Note that here, $\mathbf{k}$ can be interpreted as the internal momentum of a Cooper pair, while $\mathbf{q}$ corresponds to the external momentum. One can then perform the path integral over the Grassmann fields $\overline{\boldsymbol{\Psi}}, \boldsymbol{\Psi}$, which gives,

$$\mathcal{Z} = \int \mathcal{D}[\overline{\Delta}, \Delta] \exp\left\{\ln\det \mathcal{G}^{-1} + \sum_{i\mathbf{q}} \overline{\Delta}_{i,\mathbf{q}} \frac{\beta L^d}{v_i} \Delta_{i,-\mathbf{q}}\right\}. \quad (29)$$

Using $\ln\det \mathbf{M} = \mathrm{tr}\ln\mathbf{M}$, this results in an effective action

$$S_{\mathrm{eff}}[\overline{\Delta}, \Delta] = -\mathrm{tr}\ln\mathcal{G}^{-1} - \sum_{i\mathbf{q}} \overline{\Delta}_{i,\mathbf{q}} \frac{\beta L^d}{v_i} \Delta_{i,-\mathbf{q}}. \quad (30)$$

Expanding this to fourth order in $\Delta$ yields,

$$S_{\mathrm{eff}}[\overline{\Delta}, \Delta] = \beta L^d \sum_{i,j,\mathbf{q}} \overline{\Delta}_{i,\mathbf{q}} \left(-\frac{\delta_{ij}}{v_i} + \Pi_{ij}^{(2)}(\mathbf{q})\right)\Delta_{j,-\mathbf{q}} \\ + \beta L^d \frac{1}{4} \sum_{i,j,m,n} \overline{\Delta}_{i,\mathbf{0}}\overline{\Delta}_{m,\mathbf{0}}\Pi_{ijmn}^{(4)}(\mathbf{0})\Delta_{j,\mathbf{0}}\Delta_{n,\mathbf{0}}, \quad (31)$$

which, comparing to Eq. (5), allows one to compute the generalized Ginzburg-Landau coefficients for completing orders as described in Results.

## Model details

**Rectangular lattice model.** Consider a rectangular lattice with hopping amplitudes $t_x > t_y > 0$, on-site interaction $U$, and nearest neighbor interaction $U'$. As discussed in the main text, breaking $SU(2)$ symmetry in a single-band model requires including a small the Zeeman splitting due to a $z$-aligned magnetic field. The single-particle dispersion reads

$$\epsilon_{\mathbf{k}\sigma} = -2(t_x \cos k_x + t_y \cos k_y) + \Delta_{\mathrm{Zeeman}}\,\mathrm{sgn}(\sigma). \quad (32)$$

When the Zeeman splitting energy $\Delta_{\mathrm{Zeeman}}$ is nonzero, there are separate spin-up and spin-down Fermi surfaces, making it difficult to define singlet/even order parameters and triplet/odd order parameters in this model. For this reason, we will consider the applied magnetic field to be zero initially, only to be adiabatically turned on before the optical pulse and adiabatically turned off after the optical

**Table 1 | Basis functions $f^i$(k) that form a complete basis for $V(\mathbf{k},\mathbf{k}') = \sum_i v_i f^i(\mathbf{k})\overline{f}^i(\mathbf{k}')$ up to nearest-neighbor pairing interactions on the rectangular lattice**

| Irrep. | Pairing | $f^i$(k) |
|---|---|---|
| A | $s_{local}$ | 1 |
| B | $p_x$ | $\sqrt{2}\sin k_x$ |
| B | $p_y$ | $\sqrt{2}\sin k_y$ |
| A | $s_{ext,x}$ | $\sqrt{2}\cos k_x$ |
| A | $s_{ext,y}$ | $\sqrt{2}\cos k_y$ |

Normalization is such that $\int \frac{d^2k}{(2\pi)^2}\overline{f}^i(\mathbf{k})f^j(\mathbf{k}) = \delta_{ij}$.

**Table 2 | Basis functions $f^i_{\alpha\beta}$ that form a complete basis for $V_{\alpha\beta\alpha'\beta'}(\mathbf{k},\mathbf{k}') = \sum_i v_i f^i_{\alpha\beta}(\mathbf{k})\overline{f}^j_{\alpha'\beta'}(\mathbf{k}')$ up to nearest-neighbor pairing interactions on the honeycomb lattice**

| Irrep. | Pairing | $f^i_{\alpha\beta}$(k) |
|---|---|---|
| $A_{1g}$ | $s$ | $\frac{1}{\sqrt{2}}\begin{pmatrix} 1 & 0 \\ 0 & 1 \end{pmatrix}$ |
| $A_{1g}$ | $s_{ext}$ | $\frac{1}{\sqrt{2}}\begin{pmatrix} 0 & f_{s-ext}(\mathbf{k}) \\ \overline{f}_{s-ext}(\mathbf{k}) & 0 \end{pmatrix}$ |
| $E_{1u}$ | $p_x$ | $\frac{1}{\sqrt{2}}\begin{pmatrix} 0 & f_{d_{x^2-y^2}}(\mathbf{k}) \\ -\overline{f}_{d_{x^2-y^2}}(\mathbf{k}) & 0 \end{pmatrix}$ |
| $E_{1u}$ | $p_y$ | $\frac{1}{\sqrt{2}}\begin{pmatrix} 0 & f_{d_{xy}}(\mathbf{k}) \\ -\overline{f}_{d_{xy}}(\mathbf{k}) & 0 \end{pmatrix}$ |
| $E_{1g}$ | $d_{x^2-y^2}$ | $\frac{1}{\sqrt{2}}\begin{pmatrix} 0 & f_{d_{x^2-y^2}}(\mathbf{k}) \\ \overline{f}_{d_{x^2-y^2}}(\mathbf{k}) & 0 \end{pmatrix}$ |
| $E_{1g}$ | $d_{xy}$ | $\frac{1}{\sqrt{2}}\begin{pmatrix} 0 & f_{d_{xy}}(\mathbf{k}) \\ \overline{f}_{d_{xy}}(\mathbf{k}) & 0 \end{pmatrix}$ |
| $B_{1u}$ | $f_1$ | $\frac{1}{\sqrt{2}}\begin{pmatrix} 1 & 0 \\ 0 & -1 \end{pmatrix}$ |
| $B_{1u}$ | $f_2$ | $\frac{1}{\sqrt{2}}\begin{pmatrix} 0 & f_{s-ext}(\mathbf{k}) \\ -\overline{f}_{s-ext}(\mathbf{k}) & 0 \end{pmatrix}$ |

$f_{s-ext}(\mathbf{k}) = \frac{1}{\sqrt{3}}e^{-i\mathbf{k}\cdot\mathbf{d}_1} + \frac{1}{\sqrt{3}}e^{-i\mathbf{k}\cdot\mathbf{d}_2} + \frac{1}{\sqrt{3}}e^{-i\mathbf{k}\cdot\mathbf{d}_3}$

$f_{d_{x^2-y^2}}(\mathbf{k}) = \sqrt{\frac{2}{3}}e^{-i\mathbf{k}\cdot\mathbf{d}_1} - \frac{1}{\sqrt{6}}e^{-i\mathbf{k}\cdot\mathbf{d}_2} - \frac{1}{\sqrt{6}}e^{-i\mathbf{k}\cdot\mathbf{d}_3}$

$f_{d_{xy}}(\mathbf{k}) = -\frac{1}{\sqrt{2}}e^{-i\mathbf{k}\cdot\mathbf{d}_2} + \frac{1}{\sqrt{2}}e^{-i\mathbf{k}\cdot\mathbf{d}_3}$

pulse. We note that a small $\Delta_{Zeeman}$ has a negligible effect on the Ginzburg-Landau free energy, but must necessarily be present to allow coupling between singlet and triplet orders. Conversely, multiband models discussed below allow for the inclusion of spin-orbit coupling, obviating a magnetic field.

The form factors $f^i(\mathbf{k})$ for this model up to nearest-neighbor interactions are tabulated in Table 1. There are three order parameters in the $A$ irrep ($s$-wave) and two in the $B$ irrep ($p$-wave). In equilibrium, the system settles into a combination of the three $s$-wave orders dictated by the relative strengths of the on-site and nearest neighbor interactions. For simplicity, we re-express these order parameters in the eigenbasis of $\Pi^{(2)}_{ij}(\mathbf{0})$ [Eq. (7)], defining $\Delta_s$ in the main text to be the order parameter with the lowest eigenvalue. We then define $|\Delta_{s_{total}}|$ in Fig. 4c to be $\sqrt{\Delta^2_{s_{local}} + \Delta^2_{s_{ext,x}} + \Delta^2_{s_{ext,y}}}$.

**Honeycomb lattice model with spin-orbit coupling.** We now consider a honeycomb lattice with Kane-Mele spin-orbit coupling[55] as well as effective on-site and nearest-neighbor attractive interactions,

$$\left[h^\sigma_{\alpha\beta}(\mathbf{k})\right] = \begin{pmatrix} \lambda v^\sigma_{SO}(\mathbf{k}) - \mu & g(\mathbf{k}) \\ g^*(\mathbf{k}) & -\lambda v^\sigma_{SO}(\mathbf{k}) - \mu \end{pmatrix}, \quad (33)$$

with nearest and next-nearest-neighbor hopping

$$g(\mathbf{k}) = -t\sum_{\mathbf{d}_i} e^{-i\mathbf{k}\cdot\mathbf{d}_i}, \quad v^\sigma_{SO}(\mathbf{k}) = \text{sgn}(\sigma)\sum_{\mathbf{a}_i} \sin(\mathbf{k}\cdot\mathbf{a}_i). \quad (34)$$

Here, $\lambda$ parameterizes the strength of spin-orbit coupling, $\mu$ denotes the chemical potential, $t$ denotes the hopping parameter, $\{\mathbf{d}_i\}$ denotes the three nearest-neighbor lattice vectors, and $\{\mathbf{a}_i\}$ denotes the three next-nearest-neighbor lattice vectors ($\mathbf{a}_1 = \mathbf{d}_2 - \mathbf{d}_3$, $\mathbf{a}_2 = \mathbf{d}_3 - \mathbf{d}_1$, $\mathbf{a}_3 = \mathbf{d}_1 - \mathbf{d}_2$). For the interaction term, we consider an on-site attraction $U$ and nearest-neighbor attraction $U'$, decomposing $V_{\alpha\beta\alpha'\beta'}(\mathbf{k},\mathbf{k}')$ as in Eq. (14) using the basis functions tabulated in Table 2. Setting $U = U'$, the $\Delta_s$ state is found to have the highest critical temperature, followed by two degenerate $p_x$ and $p_y$ states. These have a $d$-wave $\mathbf{k}$-space structure, but are odd under sublattice exchange and have angular momentum $l = \pm 1$. Chiral superpositions are abbreviated as $\frac{1}{\sqrt{2}}\left(\Delta_{p_x} \pm i\Delta_{p_y}\right) \equiv \Delta_{p\pm ip}$.

**BS mode frequency and free energy barrier between two order parameters.** The Lagrangian [Eq. (4)] results in Euler-Lagrange equations of motion

$$\left(-\Gamma^{(2)}_{ij}\partial^2_t - \Gamma^{(1)}_{ij}\partial_t\right)\Delta_j = \left[\mathcal{A}_{ij} + \mathcal{B}_{ijmn}\overline{\Delta}_m\Delta_n \right.$$
$$+ \mathcal{N}^\mu_{ij}A^\mu(t) \quad (35)$$
$$\left. + \mathcal{M}^{\mu\nu}_{ij}A^\mu(t)A^\nu(t)\right]\Delta_j,$$

where $\mathcal{N}^\mu_{ij} \equiv i\frac{2e}{\hbar}\mathcal{C}^\mu_{ij}$ and $\mathcal{M}^{\mu\nu}_{ij} \equiv (i\frac{2e}{\hbar})^2\mathcal{D}^{\mu\nu}_{ij}$ are defined to absorb factors of $i\frac{2e}{\hbar}$ from minimally substituting $\nabla^\mu \longrightarrow \nabla^\mu + i\frac{2e}{\hbar}A^\mu(t)$ into Eq. (5) and assuming spatially-homogeneous order parameters and irradiation.

Focusing on the simplest case of two competing order parameters that are only coupled at linear order in $A^\mu(t)$ (e.g. $s$-wave and $p_x$-wave), the free energy for two order parameters can be written concisely as,

$$\beta\mathcal{F}^{(2)} = a_1|\Delta_1|^2 + \frac{1}{2}b_1|\Delta_1|^4 + a_2|\Delta_2|^2 + \frac{1}{2}b_2|\Delta_2|^4$$
$$+ \frac{1}{2}b_{12}\left(4|\Delta_1|^2|\Delta_2|^2 + \overline{\Delta}^2_1\Delta^2_2 + \overline{\Delta}^2_2\Delta^2_1\right), \quad (36)$$

where $a_i \equiv \mathcal{A}_{ii}$, $b_i \equiv \mathcal{B}_{iiii}$, and $b_{12}$ equals any fourth-order coefficient coupling two $\Delta_1$'s and two $\Delta_2$'s (these are all equivalent in this case). This is in agreement with Ref. 72. Assuming the system is initially in the equilibrium state, $\Delta^{(eq)}_1 = \sqrt{-a_1/b_1}, \Delta^{(eq)}_2 = 0$ (assume overall phase equals zero without loss of generality), the equation of motion for $\Delta_2$ up to linear order in $\Delta_2$ and $A^\mu(t)$ is given by,

$$(-\gamma_2\partial^2_t - \eta_2\partial_t)\Delta_2(t) = M^2\Delta_2(t) + A^\mu(t)\mathcal{N}^\mu_{21}\Delta^{(eq)}_1$$
$$+ \mathcal{O}(|\mathbf{A}(t)|^2, |\Delta_2(t)|^3), \quad (37)$$

where $\gamma_2 \equiv \Gamma^{(2)}_{22}$, $\eta_2 \equiv \Gamma^{(1)}_{22}$ (we assume the inertial and damping coefficients are diagonal), and $M^2$ is given by,

$$M^2 = a_2 - 3a_1\frac{b_{12}}{b_1} \equiv \gamma_2\Omega^2. \quad (38)$$

One can formally integrate this equation of motion as,

$$\Delta_2(t) = \mathcal{N}^\mu_{21}\Delta^{(eq)}_1 \int \frac{d\omega}{2\pi} \frac{e^{i\omega t}}{\gamma_2\omega^2 - M^2 - i\eta_2\omega}\widetilde{A}^\mu(\omega), \quad (39)$$

which yields a resonant response when $\omega^2 = \Omega^2 = M^2/\gamma_2$. We identify this as the BS mode frequency $\Omega_{BS}$. One can also derive from Eq. (36) the free energy barrier between the two states by finding the saddle point surrounded by the $\Delta_1$ minimum, the $\Delta_2$ minimum, and the $\Delta_{1,2} = 0$ maximum. The free energy difference between this saddle point and

the global minimum is given by,

$$E_{\text{barrier}} = \frac{a_1^2 b_2 + a_2^2 b_1 - 2a_1 a_2 b_{12}(2 + \cos\phi)}{(9 + 8\cos\phi + \cos 2\phi)b_{12}^2 - 2b_1 b_2} + \frac{a_1^2}{2b_1}, \qquad (40)$$

where $\phi \equiv \arg(\Delta_2/\Delta_1)$ is the relative phase between the two order parameters. For successful switching, the time-integrated pulse must supply enough kinetic energy $\sim \frac{1}{2}\gamma_2|\partial_t\Delta_2|^2$ to overcome this free energy barrier. Since the matrix element $\mathcal{N}_{ij}^\mu \propto \mathcal{C}_{ij}^\mu$ depends linearly on the spin-orbit coupling strength $\lambda$ (for small $\lambda$), we expect the requisite fluence for switching to scale as $\sim (t/\lambda)^2$ (where $t$ is the nearest-neighbor hopping amplitude).

$\mathcal{C}_{ij}^\mu$ **in terms of the Fermi surface Berry connection.** Assuming the chemical potential exists in a band with Bloch functions $|u_{\mathbf{k}}\rangle$, and dispersion $\xi_{\mathbf{k}}$ (measured with respect to the chemical potential) with a large gap $\Delta_{\text{gap}}$ to all other bands that we formally take to infinity,

$$\mathcal{C}_{ij}^\mu = \frac{i\lambda}{2}\int \frac{d^2\mathbf{k}}{(2\pi)^2}\frac{\tanh\frac{\beta\xi_{\mathbf{k}}}{2}}{2\xi_{\mathbf{k}}}\partial_\lambda\Lambda_{\mathbf{k},ij}^\mu\Big|_{\lambda\to 0} + \mathcal{O}\left(\lambda^2, \Delta_{\text{gap}}^{-1}\right) \qquad (41)$$

$$\Lambda_{\mathbf{k},ij}^\mu = \langle u_{\mathbf{k}}|\mathbf{f}_i^\dagger|u_{\mathbf{k}}\rangle\left(\langle\partial^\mu u_{\mathbf{k}}|\mathbf{f}_j|u_{\mathbf{k}}\rangle - \langle u_{\mathbf{k}}|\mathbf{f}_j|\partial^\mu u_{\mathbf{k}}\rangle\right)$$
$$- \left(\langle\partial^\mu u_{\mathbf{k}}|\mathbf{f}_i^\dagger|u_{\mathbf{k}}\rangle - \langle u_{\mathbf{k}}|\mathbf{f}_i^\dagger|\partial^\mu u_{\mathbf{k}}\rangle\right)\langle u_{\mathbf{k}}|\mathbf{f}_j|u_{\mathbf{k}}\rangle \qquad (42)$$

In the $\beta \to \infty$ limit, the integrand diverges at the Fermi surface, meaning the dominant contribution to $\mathcal{C}_{ij}^\mu$ at low temperatures and low spin-orbit strengths comes from the geometry of the Bloch states at the Fermi surface.

## Data availability
Data sets are available from the corresponding author on request.

## Code availability
Codes are available from the corresponding author on request.

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

## Acknowledgements

We thank Gene Mele and Dante Kennes for their helpful discussions. S.G. is supported by the NSF Graduate Research Fellowship Program under Grant No. DGE-1845298. C.S.W. is supported by the Deutsche Forschungsgemeinschaft (DFG, German Research Foundation) via RTG 1995. M.C. acknowledges support from the NSF under Grant No. DMR-2132591.

## Author contributions

S.G. developed the theoretical description and performed numerical calculations for the effective action approach. S.G. and C.S.W. performed numerical calculations for the microscopic approach. All authors contributed to writing the manuscript. M.C. designed and supervised the project.

## Competing interests

The authors declare no competing interests.
