## [Peer Review File · Nature Communications]

Light-induced switching between singlet and triplet superconducting statesREVIEWER COMMENTS

Reviewer #1 (Remarks to the Author):

Gassner et al. proposed a scheme to switch a singlet superconductor to a metastable triplet state using a light pulse. The linear coupling to a triplet order parameter is enabled by spin-orbit interaction. The proposal is interesting. Unfortunately, the physics behind this proposal and its experimental feasibility is not clearly demonstrated in the current manuscript. In the first example (Fig. 4), the coupling enabled by a Zeeman magnetic field looks a little too hard to realize. The authors have to turn on the magnetic field just before the optical pulse and turn it off after it (line 457~460). Therefore, I will focus on the second example: the more realistic one shown in Fig. 5. I hope the authors could address several unclear points there:

1, The c_{sp} in Eq. 10 is the key coupling that enables the light induced triplet order parameter. Given certain fermi surface parameters, an expression of c_{sp} in terms of the spin orbit coupling λ would be helpful.

2, The strength of the pump field needed to achieve the switching is missing as a function of λ and other parameters. With the c_{sp} above, one should be able to estimate it.

3, Line 312: The spin orbit strength of 0.1 eV looks too large. The corresponding value in the Kane-Mele Hamiltonian of graphene is a thousand times smaller. If the authors think the Moire bilayers are indeed good systems for the realization of the proposed effect, a reasonable spin orbit strength and the corresponding peak optical field strength should be estimated.

4, Line 341: Different relaxation rates of the two chiral $p_{\pm ip}$ states are assumed while they should be equal due to symmetry reasons. Does the system fail to fall into either minimum if the authors use equal values?

5, Line 376: I don't understand why the stabilization of the $p_{\pm ip}$ minima rely on fluctuations. I thought this happens already at the mean field level. Please clarify.

Reviewer #2 (Remarks to the Author):

The authors theoretically propose the switching of a conventional superconductor into a subdominant triplet channel. This switching is to be accomplished by the strong driving of a Bardasis-Schrieffer mode. A time-dependent Ginzburg-Landau theory is derived, which enables the authors to map out the phase space, where switching appears possible. They find that single-cycle pulse are necessary to achieve reliable switching.

Finally, an estimation of the necessary field strengths and frequencies are presented, which deem the proposal feasible with state-of-the-art technology.

This is a very interesting work. The fact that it could be within reach of current experiments should ensure that it is widely read within the community and will generate substantial impact on the further development of the field. I would be happy to recommend publication of this manuscript if the authors can overcome what I perceive could be fundamental weaknesses of their proposal:

- excitation of quasiparticles: the authors demonstrate in Fig. 4d that switching is feasible only for few-cycle or even single-cycle pulses. In this regime, the broad bandwidth of said pulses will have a substantial overlap with the quasiparticle continuum. It would therefore seem reasonable that a large (perhaps the largest) part of the photonic energy leads to the excitation of quasiparticles, rather than the strong driving of the BS mode. If am not not mistaken, this effect is not accounted for in the current simulations.

- related to this, the authors assume a large p-wave amplitude in their simulations. I assume this is necessary to create a pronounced local free energy minimum, e.g. in Fig. 4b, and thereby allow for a finite region in the parameter space in Fig. 4d, where the switching becomes possible. In this situation, would the BS mode not approach the quasiparticle continuum?

This would imply that there is a fundamental trade-off: we want the sub-dominant pairing amplitude to be large, but if it becomes too large, the quasiparticle excitation will block the switching. I think it would greatly strengthen the paper, if this trade-off was discussed.

- The parameter region in Fig. 4d is reminiscent of fractal structures, which are typical of nonlinear dynamical systems as in the present case. Is the present phase space region converged in the sense that the light blue region is really a finite area? Or does it merely appear that way as a consequence of the finite resolution?

In addition, I have a few minor comments which I ask the authors to comment on:

- Why is the value of beta changed in Fig. 2e, from 100 (line 142) to 10 (line 164)?

- Line 250: what happens in the regime above the critical temperature of the p-wave, but still below that of the s-wave? Does the metastable state vanish? How can one see this from the TDGL?

- Line 264: Why do both s- and p-wave order remain finite?

- Line 337: I do not understand the statement "We note that this free energy surface assumes Δ_{p-ip} is always zero, which while not rigorously true still provides a useful illustration for designing a switching protocol.". Why is this assumption justified?

Light-Induced Switching between Singlet and Triplet Superconducting States

Steven Gassner, Clara S. Weber, Martin Claassen

Reviewer Responses

NCOMMS-23-28464

Reviewer #1

Gassner et al. proposed a scheme to switch a singlet superconductor to a metastable triplet state using a light pulse. The linear coupling to a triplet order parameter is enabled by spin-orbit interaction. The proposal is interesting. Unfortunately, the physics behind this proposal and its experimental feasibility is not clearly demonstrated in the current manuscript. In the first example (Fig. 4), the coupling enabled by a Zeeman magnetic field looks a little too hard to realize. The authors have to turn on the magnetic field just before the optical pulse and turn it off after it (line 457~460). Therefore, I will focus on the second example: the more realistic one shown in Fig. 5. I hope the authors could address several unclear points there:

1. *The c_{sp} in Eq. 10 is the key coupling that enables the light induced triplet order parameter. Given certain fermi surface parameters, an expression of c_{sp} in terms of the spin orbit coupling λ would be helpful.*

We have now included a more detailed discussion on the calculation of c_{sp} in the main text for the honeycomb model. We perform the Matsubara summation of $\Pi_{ij}^{(2)}(\mathbf{q})$ and expand to first order in \mathbf{q} , which corresponds to the first-order gradient coefficient in the Ginzburg-Landau free energy. Remarkably, for our honeycomb model, we find that up to first order in the spin-orbit strength λ , it involves a Brillouin-zone integral of the non-Abelian Berry connection multiplied by the form factors corresponding to the s -wave and p -wave order parameters. We also now show in Methods that for low temperatures it should be primarily controlled by the Fermi surface Berry connection. We note that this constitutes a new quantum-geometric contribution to dynamics in superconductors, distinct from the quantum-geometric superfluid corrections.

2. *The strength of the pump field needed to achieve the switching is missing as a function of λ and other parameters. With the c_{sp} above, one should be able to estimate it.*

We now include in Methods a discussion of the free energy barrier between the global minimum and the metastable minimum, expressed in terms of the free energy coefficients of the Ginzburg Landau theory. Note that the *fluence* (rather than field strength) is a more useful metric for switching, since the pulse must break the dynamical inversion symmetry described in Eq. (1); this necessarily involves not just the field strength, but the time-dependence of the pulse as well. However, thanks to the reviewer's suggestion, we now argue in Methods that the requisite fluence for switching should scale as $\sim(\lambda/t)^{-2}$. For the honeycomb model, where we assume λ/t is 0.1, we now include specific magnitudes of fluence (in units of mJ/cm²) in addition to the magnitude of the field strength (in units of kV/cm) in Figure 5.

3. *Line 312: The spin orbit strength of 0.1 eV looks too large. The corresponding value in the Kane-Mele Hamiltonian of graphene is a thousand times smaller. If the authors think the Moire bilayers are indeed good systems for the realization of the proposed effect, a reasonable spin orbit strength and the corresponding peak optical field strength should be estimated.*

We thank the reviewer for this comment, since we did not originally cite many examples of strongly spin-orbit-coupled superconductors aside from the moiré platforms. We now cite more examples of candidate systems for which our honeycomb model closely matches the low-energy physics. For instance, few-layer stanene (Ref. [56-57]) is predicted to have a spin-orbit induced gap on the order of 0.1 eV. In addition, moiré heterostructures of TMDs offer generously high ratios between the spin-orbit strength and the nearest-neighbor hopping amplitude (i.e. λ/t). For instance, in Ref. [65] reports experimental data on a simulated Kane-Mele model where λ/t is of order 1. Our results in Figure 5 (which assume a ratio $\lambda/t = 0.1$) predict optical field strengths (~ 5 kV/cm) and fluences (~ 1 mJ/cm²) that lie comfortably within the limits of current experiments. Furthermore, since we now emphasize that the key coupling parameter (c_{sp}) scales with λ/t (thanks to the reviewer's suggestion), we predict that the necessary fluences required for moiré systems should only be easier to attain.

4. *Line 341: Different relaxation rates of the two chiral $p \pm ip$ states are assumed while they should be equal due to symmetry reasons. Does the system fail to fall into either minimum if the authors use equal values?*

We thank the reviewer for this point, which we now expand on in the main text. Without unequal relaxation rates, the system evolves with an equal superposition between $p + ip$ and $p - ip$ states due to an unbroken time-reversal symmetry in our effective Lagrangian. In a real system, we expect the gauge field to also couple to the bath degrees of freedom that are providing the dissipation, providing a time-reversal breaking perturbation to the relaxation rates of these states. Microscopic treatment of this mechanism is the subject of future work; for now, we introduce this difference in relaxation phenomenologically.

5. *Line 376: I don't understand why the stabilization of the $p \pm ip$ minima rely on fluctuations. I thought this happens already at the mean field level. Please clarify.*

We thank the reviewer for this clarifying question, since the explanation was ambiguously worded in the manuscript. Indeed, at mean-field level, the $p \pm ip$ states are solutions to the self-consistent gap equation, and hence are *stationary points* in the free energy. However, we have observed that a naïve quasiparticle mean-field theory treatment alone does not account for the *metastability* of these states, requiring either the higher-order Ginzburg Landau expansion performed in this work, or the inclusion of fluctuations in the quasiparticle theory. We have removed the conflicting technical wording, and mention developing a minimal set of conditions for metastability as a fundamental question that we would like to pursue in future work.

Reviewer #2

The authors theoretically propose the switching of a conventional superconductor into a subdominant triplet channel. This switching is to be accomplished by the strong driving of a Bardasis-Schrieffer mode. A time-dependent Ginzburg-Landau theory is derived, which enables the authors to map out the phase space, where switching appears possible. They find that single-cycle pulses are necessary to achieve reliable switching.

Finally, an estimation of the necessary field strengths and frequencies are presented, which deem the proposal feasible with state-of-the-art technology.

This is a very interesting work. The fact that it could be within reach of current experiments should ensure that it is widely read within the community and will generate substantial impact on the further development of the field. I would be happy to recommend publication of this manuscript if the authors can overcome what I perceive could be fundamental weaknesses of their proposal:

- excitation of quasiparticles: the authors demonstrate in Fig. 4d that switching is feasible only for few-cycle or even single-cycle pulses. In this regime, the broad bandwidth of said pulses will have a substantial overlap with the quasiparticle continuum. It would therefore seem reasonable that a large (perhaps the largest) part of the photonic energy leads to the excitation of quasiparticles, rather than the strong driving of the BS mode. If am not not mistaken, this effect is not accounted for in the current simulations.

We thank the reviewer for raising this important point. Our revised manuscript now takes careful consideration of this issue, particularly in the honeycomb lattice results, where we now discuss an intriguing tradeoff between the need for dynamical inv. sym. breaking (favoring short pulses) while minimizing the excitation of quasiparticles (favoring long pulses). In brief, we find a healthy range of pulse widths that still lead to efficient switching when we center the Gaussian pulse at slightly *less* than the BS mode frequency, while ensuring negligible overlap with the quasiparticle continuum. To illustrate this important point, we have substantially revised Figures 4 and 5, which now include the revised pulse parameters together with plots emphasizing the location of the BS mode relative to the quasiparticle continuum and the spectral profile of the pulse.

- related to this, the authors assume a large p -wave amplitude in their simulations. I assume this is necessary to create a pronounced local free energy minimum, e.g. in Fig. 4b, and thereby allow for a finite region in the parameter space in Fig. 4d, where the switching becomes possible. In this situation, would the BS mode not approach the quasiparticle continuum?

This would imply that there is a fundamental trade-off: we want the sub-dominant pairing amplitude to be large, but if it becomes too large, the quasiparticle excitation will block the switching. I think it would greatly strengthen the paper, if this trade-off was discussed.

Fortunately, there is in fact no tradeoff of this sort, since increasing the sub-dominant pairing amplitude actually *decreases* the BS mode frequency. One can prove this by combining Eq. (6) (which says that a_p scales as $\frac{1}{|v_p|}$) with Eq. (11) (which says that $\Omega_{BS} \sim a_p^{1/2} \sim |v_p|^{-1/2}$). Intuitively, increasing the pairing amplitude causes the sub-dominant phase to become lower in free energy, eventually causing it to be the new global minimum. But before this point, understanding the BS mode as the curvature of the free energy in the direction of the sub-dominant minimum, this curvature becomes *shallower* as the sub-dominant minimum is brought lower in free energy.

- The parameter region in Fig. 4d is reminiscent of fractal structures, which are typical of nonlinear dynamical systems as in the present case. Is the present phase space region converged in the sense that the light blue region is really a finite area? Or does it merely appear that way as a consequence of the finite resolution?

We thank the reviewer for clarifying this. Indeed, the light blue region has a truly finite area. Changing the input parameters by very small (including irrational) amounts does not lead to different final states due to dissipation: provided that the pulse nudges the order parameter sufficiently close to the metastable minimum, damping η ensures that the system remains trapped in the metastable phase. This is also true for our new Figure 5e, which pertains to our honeycomb model.

In addition, I have a few minor comments which I ask the authors to comment on:

- Why is the value of beta changed in Fig. 2e, from 100 (line 142) to 10 (line 164)?

We thank the reviewer for pointing this out. There was no particular reason for an inconsistent choice of parameters, so we have now fixed Figure 2 to have a consistent temperature and pairing strengths across all subfigures.

- Line 250: what happens in the regime above the critical temperature of the p -wave, but still below that of the s -wave? Does the metastable state vanish? How can one see this from the TDGL?

Indeed, the metastable state does vanish. Above the critical temperature of the p -wave state, there is no p -wave solution (i.e. p -wave stationary point in the free energy) to consider. One can see this from the TDGL by setting all other Δ_i to zero, and seeing that when $a_p > 0$, the only stationary point as a function of Δ_p would be $\Delta_p = 0$.

- Line 264: Why do both s - and p -wave order remain finite?

We reworded this line to be clearer. This is simply a plotting consideration, allowing visualization of a two-dimensional *real* parameter space instead of a two-dimensional *complex* parameter space. Since the relative phase of Δ_s and Δ_p remains zero throughout the entire time evolution, one can plot the free energy surface as though they are both *real* order parameters. This is not meant to claim that they are always both finite.

- Line 337: I do not understand the statement "We note that this free energy surface assumes Δ_{p-ip} is always zero, which while not rigorously true still provides a useful illustration for designing a switching protocol.". Why is this assumption justified?

We thank the reviewer for pointing out this misleading wording. This is again a plotting consideration, not an assumption being used in the theory. To be precise, the full trajectory of the order parameter exists in a higher dimensional free energy landscape spanned by the order parameters Δ_s , Δ_{p+ip} , and Δ_{p-ip} , which would require at least three dimensions to fully visualize. Line 337 is simply meant to clarify that the 2D free energy surface in Figure 5 is taken along the "slice" $\Delta_{p-ip} = 0$. Even though the order parameter trajectory briefly leaves this

subspace during the pump pulse, this is still a useful illustration for designing a switching protocol.

Summary of changes to the manuscript (highlighted in red in the manuscript)

Line 46: Added “provided that long-range Coulomb interactions do not push the mode into the pair-breaking continuum [Hackner2023]”

Line 142: Changed parameters in Fig 2 to be consistent across all subfigures, and therefore stated “For numerical expediency, in Fig 2 we use effective interactions $U = U' = -2$ and $\beta=10$, in units of hopping.”

FIG. 2: Updated the graphic for Fig 2 so that the choice of interaction parameters and temperature are consistent across all subfigures.

Lines 242 and 249: After updating Fig 4 based on the reviewers’ suggestions, added the new parameters to the main text: “chemical potential $\mu=-0.1$, and interaction parameters $v_s = -1.75$, $v_{px}=v_{py}=-2.5$, in units of hopping...the phenomenological inertial/damping coefficients are set to $\gamma = 1.0$, $\eta = 0.1$.”

Line 263: Clarified this sentence in response to Reviewer #2: “In the rectangular model, this coupling keeps the s- and px-wave order parameters completely real, allowing the dynamics to be completely captured by a visualizable two-dimensional free energy contour plot”

FIG. 4: Updated the graphic with new results that respect the quasiparticle considerations raised by Reviewer #2. Here, driving with few-cycle pulses at a frequency slightly lower than the BS mode frequency allows successful switching while avoiding spectral overlap with the pair-breaking continuum. Fig 4e illustrates this. Caption is changed to reflect this.

Line 276: Changed wording from “resonant strong driving at” to “strong driving near” the Bardasis-Schrieffer mode frequency, to reflect the new strategy that will be described later in this paragraph.

Line 285: Expanded discussion given the improved FIG. 4. “Figures 4d and 4e together reveal an intriguing trade-off whereby the pulse needs to be short enough to allow for strong dynamical symmetry breaking while being wide enough to avoid excitation of quasiparticles across the gap (excitations that are not captured by TDGL). The latter constraint leads us to detune the central frequency of the Gaussian pulse to be slightly below Ω_{BS} , decreasing the spectral overlap with the quasiparticle continuum while still maintaining a large overlap with the BS mode.”

Line 298: Added citations to a larger variety of real materials whose low-energy physics are captured by our honeycomb model. “This models the low-energy physics near the Dirac points of several systems of recent interest, including honeycomb superconductors [Liao2018, Zhao2020], kagome superconductors [Mielke2021, Shi2022, Chakraborty2023], and many moiré heterostructures of transition metal dichalcogenides [Naik2018, Xian2021, Angeli2021, Kennes2021, Zhao2022].”

Line 312: Included an explicit expression of the key coupling parameter “ C^{μ}_{ij} ” in terms of the spin-orbit coupling for the honeycomb model, in response to Reviewer #1. The result is revealed to depend strongly on the quantum geometry of the bands near the Fermi energy, depending explicitly on the non-Abelian Berry connection weighted by the form factors.

FIG. 5: Updated graphic analogously to FIG. 4. Added both Fig 5d (showing spectral overlap with Ω_{BS} and the quasiparticle continuum) and 5e (showing a parametric plot analogous to 4e, with axes in physical units). Updated caption to reflect the new subfigures.

Line 344: Added note referring to the new subfigure Fig 5e: “Figure 5e reveals a broad region of pulse widths (~ 0.1 ps) and fluences (~ 1.0 mJ/cm²) that lead to successful switching from s to a chiral triplet p+ip state via an ultrafast circularly polarized pulse.”

Line 354: Added a clarifying note in response to Reviewer #1, in reference to the difference in relaxation parameters: “Though we introduce this as a phenomenological parameter, one can expect this to arise microscopically from pump-induced time-reversal symmetry breaking of the bath degrees of freedom that provide the dissipation, which is not captured by TDGL theory.”

Line 394: Added a direction of future work inspired by the arXiv work suggested by the Editor: “Lastly, throughout this work we use effective renormalized interaction strengths v_i as phenomenological parameters. Detailed modeling of these parameters in candidate materials, starting from screened Coulomb interactions and incorporating RPA effects [Hackner2023], is an important direction for future work.”

Line 485: After updating Fig 4, we deemed it clearer to *remove* Fig 6 from the methods and instead plot in Fig 4c a “total” s-wave order parameter totaling the fluctuations in all order parameters in the “A” irrep (these include local s-wave pairing and extended s-wave pairing). In the original manuscript, we omitted the extended s-wave parameters in Fig 4 and included all of them in Fig 6 in the Methods. In this line, we specify the precise meaning of this “total” s-wave order parameter: “There are three order parameters in the A irrep (s-wave) and two in the B irrep (p-wave). In equilibrium, the system settles into a combination of the three s-wave orders dictated by the relative strengths of the on-site and nearest-neighbor interactions. For simplicity, we re-express these order parameters in the eigenbasis of $\Pi_{ij}^{(2)}(\mathbf{0})$ [Eq. (7)], defining Δ_s in the main text to be the order parameter with the lowest eigenvalue. We then

define $|\Delta_{s_{total}}|$ in Figure 4c to be $\sqrt{\Delta_{s-local}^2 + \Delta_{s-ext-x}^2 + \Delta_{s-ext-y}^2}$.”

Lines 513 and 536: Expanded the Methods section on “Resonant frequency between two order parameters” to include not only the derivation of the BS mode frequency, but also the free energy barrier between two competing orders, to help address one of Reviewer #1’s comments.

Line 550: Added a Methods section on expression the key coupling parameter C_{ij}^{μ} in terms of the Fermi surface Berry connection, to help address one of Reviewer #1’s comments.

Line 576: Data and Code availability statement added. All our code is written in Mathematica (no custom packages) and is available on reasonable request.

References added:

[39] Nico A. Hackner and P. M. R. Brydon, “Bardasis-Schrieffer-like phase mode in a superconducting bilayer,” (2023), arXiv:2306.16611 [cond-mat.supr-con].

[56] C. Mielke, Y. Qin, J.-X. Yin, H. Nakamura, D. Das, K. Guo, R. Khasanov, J. Chang, Z. Q. Wang, S. Jia, S.

- Nakatsuji, A. Amato, H. Luetkens, G. Xu, M. Z. Hasan, and Z. Guguchia, "Nodeless kagome superconductivity in LaRu₃Si₂," *Phys. Rev. Mater.* 5, 034803 (2021).
- [57] Mengzhu Shi, Fanghang Yu, Ye Yang, Fanbao Meng, Bin Lei, Yang Luo, Zhe Sun, Junfeng He, Rui Wang, Zhicheng Jiang, Zhengtai Liu, Dawei Shen, Tao Wu, Zhenyu Wang, Ziji Xiang, Jianjun Ying, and Xianhui Chen, "A new class of bilayer kagome lattice compounds with Dirac nodal lines and pressure-induced superconductivity," *Nature Communications* 13, 2773795 (2022).
- [58] S. Chakraborty, Ram Kumar, and N. Mohapatra, "Effect of tunable spin-orbit coupling on the superconducting properties of LaRu₃Si₂ containing kagome-honeycomb layers," *Phys. Rev. B* 107, 024503 (2023).
- [59] Yong Xu, Binghai Yan, Hai-Jun Zhang, Jing Wang, Gang Xu, Peizhe Tang, Wenhui Duan, and Shou-Cheng Zhang, "Large-gap quantum spin hall insulators in tin films," *Phys. Rev. Lett.* 111, 136804 (2013).
- [60] Menghan Liao, Yunyi Zang, Zhaoyong Guan, Haiwei Li, Yan Gong, Kejing Zhu, Xiao-Peng Hu, Ding Zhang, Yong Xu, Ya-Yu Wang, Ke He, Xu-Cun Ma, Shou-Cheng Zhang, and Qi-Kun Xue, "Superconductivity in few-layer stanene," *Nature Physics* 14, 344–348 (2018).
- [61] Chen-Xiao Zhao and Jin-Feng Jia, "Stanene: A good platform for topological insulator and topological superconductor," *Frontiers of Physics* 15, 53201 (2020).
- [67] Sebastiano Peotta and Päivi Törmä, "Superfluidity in topologically nontrivial flat bands," *Nature Comm.* 6, 8944 (2015).
- [68] Paivi Torma, Sebastiano Peotta, and Bogdan A. Bernevig, "Superconductivity, superfluidity and quantum geometry in twisted multilayer systems," *Nature Rev. Phys.* 4, 528–542 (2022).

REVIEWER COMMENTS

Reviewer #1 (Remarks to the Author):

My concerns are addressed in the revised manuscript and in the reply. I recommend the publication of the revised manuscript after the typo in Fig. 5c (both legends appear as p+ip) is corrected.

Reviewer #2 (Remarks to the Author):

The authors have addressed all the points raised by the first round of reports. They have substantially revised their manuscript, and responded convincingly to most of these points.

I am however not quite satisfied with their discussion of the quasiparticle excitation. In their response, they write "In brief, we find a healthy range of pulse widths that still lead to efficient switching [..], while ensuring negligible overlap with the quasiparticle continuum". While this is wonderful news, I cannot understand how this insight is gained from the revised manuscript, because the discussion is purely qualitative. In particular,

- How is this 'healthy range' identified? According to which measures? Surely, there should be a tradeoff between photon fluence and acceptable overlap with the continuum.
- What is a 'slight detuning'?

A more quantitative analysis, e.g., by estimating the heating rate from linear absorption at the fluence range employed in the two figures would be much more convincing. I understand that a long discussion would detract from the main message, but some numbers should be given, e.g. in the Methods section.

Once this remaining issue is resolved, I believe the manuscript will be appropriate for publication in Nature Communications.

Light-Induced Switching between Singlet and Triplet Superconducting States

Steven Gassner, Clara S. Weber, Martin Claassen

Reviewer Responses

NCOMMS-23-28464

Reviewer #1

My concerns are addressed in the revised manuscript and in the reply. I recommend the publication of the revised manuscript after the typo in Fig. 5c (both legends appear as $p+ip$) is corrected.

We thank the reviewer for spotting that typo, which is now fixed. We are grateful for their recommendation.

Reviewer #2

The authors have addressed all the points raised by the first round of reports. They have substantially revised their manuscript, and responded convincingly to most of these points.

I am however not quite satisfied with their discussion of the quasiparticle excitation. In their response, they write "In brief, we find a healthy range of pulse widths that still lead to efficient switching [..], while ensuring negligible overlap with the quasiparticle continuum". While this is wonderful news, I cannot understand how this insight is gained from the revised manuscript, because the discussion is purely qualitative. In particular,

- How is this 'healthy range' identified? According to which measures? Surely, there should be a tradeoff between photon fluence and acceptable overlap with the continuum.*
- What is a 'slight detuning'?*

A more quantitative analysis, e.g., by estimating the heating rate from linear absorption at the fluence range employed in the two figures would be much more convincing. I understand that a long discussion would detract from the main message, but some numbers should be given, e.g. in the Methods section.

Once this remaining issue is resolved, I believe the manuscript will be appropriate for publication in Nature Communications.

We thank the reviewer for this comment. Per their suggestion, we have now added supplementary material that provides a more quantitative analysis of resonances with quasiparticle excitations. We include supplementary plots that, given the window of pulse parameters presented in Figures 4e and 5e, explicitly show the fraction of the pump fluence coming from frequencies above the equilibrium superconducting gap. This makes more explicit the tradeoff between photon fluence and the excitation of quasiparticles, which was absent in our previous manuscript. Of course, it would be useful to reliably estimate a microscopic heating rate from this excitation fluence, and we considered a few approaches. The linear absorptive conductivity would underestimate such a heating rate, both because our driving schemes are beyond the linear response regime and because the linear absorptive optical conductivity is known to be very small in multiband superconductors without disorder. On the other hand, a

microscopic approach of tracking non-equilibrium quasiparticle dynamics in the singlet-to-triplet-pairing switching regime necessitates including a microscopic model for dissipation (that permits the order parameter to relax into the triplet minimum and in turn permits photoexcited quasiparticles to relax), which is beyond the scope of this manuscript. We do, however, find this to be an important question for future work, which we now mention in the Summary and Outlook.

REVIEWERS' COMMENTS

Reviewer #2 (Remarks to the Author):

My remaining concern is addressed and I am happy to recommend the publication of the manuscript.